# Wat zei je?
# Detecting Out-of-Distribution Translations with Variational Transformers

## Abstract

We detect out-of-training-distribution sentences in Neural Machine Translation using the Bayesian Deep Learning equivalent of Transformer models. For this we develop a new measure of uncertainty designed specifically for long sequences of discrete random variables—i.e. words in the output sentence. Our new measure of uncertainty solves a major intractability in the naive application of existing approaches on long sentences. We use our new measure on a Transformer model trained with dropout approximate inference. On the task of German-English translation using WMT13 and Europarl, we show that with dropout uncertainty our measure is able to identify when Dutch source sentences, sentences which use the same word types as German, are given to the model instead of German.

## 1 Introduction

Statistical Machine Translation (SMT, (Brown et al., 1993; Och, 2003)), built on top of probabilistic modelling foundations such as the IBM alignment models (Vogel et al., 1996; Brown et al., 1993; Gal & Blunsom, 2013), has largely been replaced in recent years following the emergence of Neural Machine Translation approaches (NMT, (Kalchbrenner & Blunsom, 2013; Bahdanau et al., 2015; Luong et al., 2015; Vaswani et al., 2017)). This change has brought with it huge performance gains to the field (Sennrich et al., 2016), but at the same time we have lost many desirable properties of these models. Statistical probabilistic models can inform us when they are guessing at random on inputs they never saw before (Ghahramani, 2015). This information can be used, for example, to detect out-of-training-distribution examples for selective classification by referring uncertain inputs to an expert for annotation (Leibig et al., 2017), or for a human-in-the-loop approach to reduce data labelling costs (Gal et al., 2017; Walmsley et al., 2019; Kirsch et al., 2019).

With new tools in machine learning we can now incorporate such probabilistic foundations into deep learning NLP models without sacrificing performance. This field, known as Bayesian Deep Learning (BDL, (Neal, 2012; Gal, 2016)), is concerned with the development of scalable tools which capture *epistemic* uncertainty—the model's notion of "I don't know", a measure of a model's lack of knowledge e.g. due to lack of training data, or when an input is given to the model which is very dissimilar to what the model has seen before. Such BDL tools have been used extensively in the Computer Vision literature (Kendall & Gal, 2017; Litjens et al., 2017), and have been demonstrated to be of practical use for applications including medical imaging (Litjens et al., 2017; Nair et al., 2020), robotics (Gal et al., 2016; Chua et al., 2018), and astronomy (Hon et al., 2018; Soboczenski et al., 2018; Hezaveh et al., 2017).

In this paper we extend these tools, often used for vision tasks, to the language domain. We demonstrate how these tools can be used effectively on the task of selective classification in NMT by identifying source sentences the translation model has never seen before, and referring such source sentences to an expert for translation. We demonstrate this with state-of-the-art Transformer models, and show how model performance increases when rejecting sentences the model is uncertain about—i.e. the model's measure of epistemic uncertainty correlates with mistranslations.

For this we develop several measures of epistemic uncertainty for applications in natural language processing, concentrating on the task of machine translation (§3). We compare these measures both with standard deterministic Transformer models, as well as with Variational Transformers, a new approach we introduce to capture epistemic uncertainty in sequence models using dropout approximate inference (Gal, 2016). We give an extensive analysis of the methodology, and compare

the different approaches quantitatively in the out-of-training-distribution settings (§4), which shows our proposed uncertainty estimate BLEUVar works well for measuring the epistemic uncertainty for machine translation. We also analyse the performance of BLEUVar qualitatively from both the influence of sentence length and from the linguistic perspective. We finish with a discussion in potential use cases for the new methodology proposed.

The closest NLP task to the above problem definition is the quality estimation (QE) task in Machine Translation (Specia et al., 2010; Blatz et al., 2004), which tries to solve a similar problem by predicting the quality of a translation with a score called Human Translation Error Rate (HTER, (Snover et al., 2006)). This is done by training a surrogate QE model on source sentences and their corresponding machine-generated translations in a specific domain, with the target of the surrogate to predict the the percentage of edits needed to be fixed. While many methods have been shown to successfully solve the task of estimating the quality of translations (Kim et al., 2017; Martins et al., 2016; 2017; Kreutzer et al., 2015), by definition QE crucially relies on examples of mistranslations to train the surrogate. The assumption that such training data is available is often violated in practice though (e.g. in active learning), thus existing approaches in QE research cannot generally be used to detect out-of-training-distribution examples (see Appendix C for detail discussion about the differences between QE and our task, as well as other related work that is similar to ours but not solving the same problem).

## 2 BACKGROUND: UNCERTAINTY IN DEEP LEARNING

For most machine learning models, the optimisation objectives give us a point estimate of the model parameters, which maximise the likelihood of the model generating the training data (i.e. $p(Y|X, \omega = \omega^*)$, $\omega^* \in \Omega$ s.t. $\Omega$ is the set of all possible model parameters, $Y, X$ are the training data). Such point estimate $\omega^*$ gives us a very good prediction when the test data follow the same distribution as the training data distribution. Given a new input $x^*$ at test time, the model prediction for the corresponding $y^*$ is

$$\hat{y}^* = \arg\max_{y^*} p(y^*|x^*, \omega^*). \tag{1}$$

However, we cannot expect the model to perform well on out-of-distribution (OOD) data which it never saw before. Instead, we would wish for the model to indicate its uncertainty towards such inputs. We could use $p(\hat{y}^*|x^*, \omega^*)$ as an estimate for model uncertainty, but as we show below, it would not be a well calibrated one. It might be the case that many $\omega$ might give equally good predictions on the train set, but might widely disagree with their predictions on OOD data. In fact, $\omega^*$ might give arbitrary predictions on OOD training data which is very dissimilar to previously observed inputs. Thus, a high score does not distinguish whether $x^*$ is OOD or not, and whether we should trust the model's prediction.

### 2.1 BAYESIAN INFERENCE

Bayesian probabilistic models capture the notion of uncertainty explicitly. Rather than considering a single point estimate $\omega^*$, Bayesian models aim to capture the entire distribution of $\omega$ from the training data. The resulting distribution is called **posterior** distribution

$$p(\omega|X, Y) = \frac{p(Y|X, \omega)p(\omega)}{p(Y|X)}. \tag{2}$$

At test time, we can make prediction about the corresponding $y^*$ by integrating out all possible $\omega$

$$p(y^*|x^*, X, Y) = \int p(y^*|x^*, \omega)p(\omega|X, Y)\mathrm{d}\omega. \tag{3}$$

Using the variance of the predictive distribution $p(y^*|x^*, X, Y)$ as the uncertainty measure would have taken into account the variance of $\omega$. Hence, an uncertainty measure based on this quantity could be a strong indicator for $x^*$ being OOD.

### 2.2 APPROXIMATE INFERENCE

The difficulty in doing Bayesian inference comes from the intractability of calculating the evidence

$$p(Y|X) = \int p(Y|X, \omega)p(\omega)\mathrm{d}\omega. \tag{4}$$

There might be a closed form solution for a simple model. But for most interesting problems, it is too difficult to compute an exact solution. Therefore, approximations are often used for such inference problems.

Variational inference (VI) is a pragmatic popular method for doing approximate inference (Jordan et al., 1999). The method involves defining an approximating distribution $q_\theta(\omega)$, and trying to find the parameter $\theta$ for that minimises the Kullback-Leibler (KL) divergence (Kullback & Leibler, 1951) between $q_\theta(\omega)$ and the posterior

$$\text{KL}(q_\theta(\omega)\|p(\omega|X,Y)) = \int q_\theta(\omega)\log\frac{q_\theta(\omega)}{p(\omega|X,Y)}\mathrm{d}\omega. \tag{5}$$

The resulting $\theta^*$ gives us the closest approximation for the posterior in the family of distribution $q_\theta(\omega)$. However, calculating the KL divergence here is also intractable as we still have to do the integration for the evidence in posterior.

Fortunately, a tractable and equivalent objective for minimising the KL divergence is to maximise the evidence lower bound (ELBO, (Blei et al., 2017)) of $q_\theta(\omega)$

$$\text{ELBO}(q_\theta) = \int q_\theta(\omega)\log p(Y|X,\omega)\mathrm{d}\omega - \text{KL}(q_\theta(\omega)\|p(\omega)). \tag{6}$$

This is tractable as we know all the distributions (both $q_\theta(\omega)$, $p(Y|X,\omega)$ and $p(\omega)$ are defined by the model or by our assumptions).

### 2.3 BAYESIAN INFERENCE IN DEEP LEARNING

Almost all deep models treat the units in a deep neural network as deterministic functions. To adapt Bayesian methods in deep learning, we need to first turn our model into a probabilistic model. It can be done by modelling the weights in each unit of the network as samples from probability distributions. Such networks are called Bayesian neural networks (Neal, 2012). One major challenge with Bayesian neural networks is that the integration in ELBO becomes intractable when we have more than one hidden layer (Gal, 2016).

Many works have tried tackling this problem. One practical method proposed by Gal (2016) is MC Dropout. Gal (2016) showed that optimising any neural networks with dropout can be viewed as an approximate inference in a probabilistic model (when dropout $p$ is tuned correctly), which implies that a trained neural network with dropout can be interpreted as a Bayesian neural network Gal (2016). Stochastic forward passes with dropout 'turned-on' at test time then correspond to draws from the predictive distribution. Here we extend on these ideas and propose the *Variational Transformer*, which is based on MC Dropout applied to the original Transformer model. We perform extensive empirical evaluation with this model on the task of NMT. Representing uncertainty in a translation model is the first step towards detecting OOD data. We next discuss how to *use* this uncertainty effectively, and provide the main contribution of this work.

## 3 MEASURES OF UNCERTAINTY FOR NMT

Principally, we care about measuring the variance of a model's outputs around some given input point. In the context of a simple classifier model, the solution is often found by measuring the mutual information between the predicted discrete distribution and model parameters, evaluating the output's entropy, or simply computing the variance of model outputs (Gal, 2016). In the domain of language, however, there are many semantically equivalent alternatives to the same prediction, and it is a difficult matter to measure the disagreement between the predicted discrete sequences, which in turn complicates the estimation of variance in the output space. Much worse, when attempting to naively use MI or entropy with long sequences or large sets of discrete random variables, we quickly discover that even approximate integration over the product space becomes prohibitive (Kirsch et al., 2019). In order to capture epistemic uncertainty in the task of NMT, we propose several measures of uncertainty appropriate for long sequences of discrete variables (Beam Score and Sequence Probability are measures similar to (Wang et al., 2019; Fomicheva et al., 2020)):

1. **Beam Score:** we assign a confidence to output $y$ generated (using beam search) from input $x$ using the score assigned to $y$'s beam (Wu et al., 2016), where $\text{length\_penalty}(y;\alpha) = \left(\frac{5+|y|}{5+1}\right)^\alpha$.

$$\text{BS} = \frac{\log\left(p_{\omega^*}(y|x)\right)}{\text{length\_penalty}(y;0.6)} \tag{7}$$

2. **Sequence Probability:** we assign a confidence to the deterministic model output $y$ generated from input $x$ by taking the log predictive probability under the weight distribution.

$$\text{SP} = \frac{\log\left(\mathbb{E}_{\omega\sim q_{\theta^*}(\omega)}\, p_\omega(y|x)\right)}{\text{length\_penalty}(y; 0.6)} \tag{8}$$

3. **BLEU Variance:** ideally, we would like to measure the variance of outputs $y$ as the uncertainty at an input $x$, i.e.:

$$\text{Var} = \mathbb{E}_{\omega\sim q_{\theta^*}(\omega)}\mathbb{E}_{y\sim p_\omega(y|x)}\left(y-\mu\right)^2 = \mathbb{E}_{\omega\sim q_{\theta^*}(\omega)}\mathbb{E}_{y,y'\sim p_\omega(y|x)}\frac{1}{2}\left(y-y'\right)^2, \tag{9}$$

where $\mu = \mathbb{E}_{y\sim p_\omega(y|x)}[y]$. If we treat sentences as points in some high dimensional space, $||y-y'||$ corresponds to a distance between these two points, which is a numerical value representing the difference between two sentences. Thus, any metric for measuring the difference between sentences will allow us to calculate the variance of output $y$. In our experiments we choose BLEU (Papineni et al., 2002). The BLEU score of a candidate text to the reference text is a number between 0 and 1, with the value closer to 1 indicating the two texts are more similar[1]. Now, we can estimate the variance at an input $x$ by producing pairs of outputs from the model and measuring the squared complement of the BLEU between them, i.e. $||y-y'||^2 := (1 - \text{BLEU}(y,y'))^2$, and we have:

$$\text{BLEUVar} = \mathbb{E}_{\omega\sim q_{\theta^*}(\omega)}\mathbb{E}_{y,y'\sim p_\omega(y|x)}\left(1-\text{BLEU}(y,y')\right)^2 \tag{10}$$

For the *beam score* we use the deterministic model found by gradient descent and simply take the probabilities from under its output probabilities. This will be our baseline. For *sequence probability* and *BLEU variance* we use MC Dropout (Gal, 2016) and take a number of samples ($N$) to estimate the expectations[2]:

$$\text{SP} \approx \frac{\log\left(\sum_{i=1}^N p_{\omega_i}(y|x)\right)}{\text{length\_penalty}(y; 0.6)} \tag{11}$$

For the BLEUVar approximation, we opt for decoding outputs using beam search applied to different model samples (realised by randomising the dropout masks) and measuring the complement BLEU between pairs of these examples.

$$\text{BLEUVar} \approx \sum_{i=1}^N \sum_{j\neq i}^N \left(1-\text{BLEU}(\text{dec}_{\omega_i}(x), \text{dec}_{\omega_j}(x))\right)^2. \tag{12}$$

Additionally, out of the $N$ sample sequences generated by BLEUVar, we need to choose one or generate a new sequence as the result for a specific input. In regression, the mean of the samples is normally chosen as the result, which is an approximation for the *predictive mean*. In our case, the 'mean' of $N$ sentences is hard to derive or even not properly defined. Therefore, we use the sampled sequence that is the closest to the ideal 'mean' as an approximation. Since we can measure the disagreement between any two sentences using BLEU, the sequence that is the closest to the 'mean' of the $N$ samples must have the smallest disagreement with rest of the $N-1$ samples. Hence, the final output sequence of the method BLEU Variance is:

$$\tilde{\mu} = \arg\min_{y_i}\left(\sum_{\forall j\neq i}^N \left(1-\text{BLEU}(y_i, y_j')\right) + \sum_{\forall j\neq i}^N \left(1-\text{BLEU}(y_j', y_i)\right)\right). \tag{13}$$

One remark for the above three methods is that the two methods BS and SP have the same resulting translation $y$ given $x$, but with different values as its uncertainty estimates. For the method BLEUVar, it uses $\tilde{\mu}$ for the resulting translation, and a value in a different range as the uncertainty estimate. This is reflected in the plots from the experiments section. For example, in Figure 6 (b), BS and SP converged into the same value, while BLEUVar converged to a different value.

### 3.1 EVALUATING UNCERTAINTY IN SEQUENCE MODELS

A number of uncertainty evaluation metrics have been proposed for standard classifier networks such as the popular ECE and MSE metrics (Guo et al., 2017); however, for sequence modelling these classification-specific metrics are not applicable. Instead, we opt for the *performance versus retention* curve method for evaluating our uncertainty measures following Filos et al. (2019).

---

[1]A common practice in the NMT literature is to scale it up by $\times 100$, i.e. in the range of $[0, 100]$.

[2]The approximations below should have constant scaling factors, but these don't impact our evaluation metric (performance-retention curves) so we leave them out for simplicity.

The performance-retention curve indicates how well an uncertainty measure would perform if the $k\%$ *least* certain outputs were deleted from the test dataset. The $x$-axis ranges along the fraction of data retained, while the $y$-axis measures some performance metric of the model on the retained data. A performance-retention curve of a well-calibrated uncertainty measure will see a clear and sustained improvement in performance as low-confidence predictions are excluded from the test set; while a poorly-calibrated model will yield a curve that either lacks a trend or tends to lay beneath the well-calibrated metric's curve. In addition, we also calculate the area under a curve (AUC) as a summary statistic for comparing the curves. A larger AUC corresponds to a higher curve on average.

## 4 EXPERIMENTS

Experiments consist of evaluations on both in-distribution (see Appendix A) and out-of-distribution test sets. The implementation of the Transformer architecture we use is taken from the Tensor2Tensor (Vaswani et al., 2018) repository.

As discussed in the previous sections we use performance-retention curves to evaluate the different uncertainty estimates of our models. We also use scatter plots of the uncertainty versus pairwise BLEU of the predictions to offer another gauge of model uncertainty (see Appendix A.2) To visualise the quality of uncertainty estimates for Transformers, we can order the generated sequence based on their uncertainty estimates from the most confident to the most uncertain, and plot BLEU scores as a function of the fraction of data retained starting from the least uncertain output. For a good uncertainty estimate, we are expecting to see the BLEU scores decrease when the fraction of retained data increases.

The following datasets were used in our experiments: **(1) WMT EN $\leftrightarrow$ DE**: The training set for translation tasks between English (EN) and German (DE) composed of *news-commentary-v13* with $284k$ sentences pairs, *wmt13-commoncrawl* with $2.4m$ sentences pairs and *europarl-v7* with $1.9m$ sentences pairs, in total $4.6m$ sentences pairs. The test set was the *newstest2014* with size $3k$ from *WMT 2014*. **(2) WMT NL $\rightarrow$ EN**: The test set for Dutch (NL) to English (EN) translation was a subset (size $3k$ sentences pairs) of *news-commentary-v14*.

In the experiments below, we denote the results from methods *beam score*, *sequence probability*, and *BLEU Variance* as *BS*, *SP* and *BLEUVar* respectively. If a suffix is added such as *BLEUVar-10*, then the suffix *10* indicates the number of samples taken during MC dropout.

### 4.1 OUT-OF-DISTRIBUTION EXPERIMENTS

The in-distribution tests showed in Appendix A illustrate that when the training set is large enough, the three uncertainty estimates have similar performance if the test set is in the same domain as the training set. However, when the training set only has limited data (e.g. $50k$ compares to $4.6m$), BLEUVar outperforms BS and SP even though the test set is in the same domain as the training set. One explanation is that with the limited amount of training data, the model was not able to learn a distribution that capture the data from the test set. Therefore, to some extend, the test data are slightly out-of-distribution.

In this section, we will have a look of experiments under two different obviously out-of-distribution settings (which in our application is equivalent to a *domain shift*) for evaluating our uncertainty estimates.

The first experiment exploits the fact that the wordings and the sentence patterns might vary across different content domains in the same language. For example, sentences from a legal document and sentences from social media are different in terms of formality, even though they can both be in the same language. If we train a model using data from one domain and test it with data from another domain, then such input would be out-of-distribution for the model.

The second experiment explores an extreme case of out-of-distribution setting. Given a trained model for translation task from language $A$ to language $B$, if we test it on input from language $C$ s.t. $C \neq A$, it would be an out-of-distribution input.

### 4.1.1 UNCERTAINTY CAUSED BY DIFFERENT CONTENT DOMAINS

To evaluate our uncertainty measures on out-of-content-domain test set, we trained a model using only the *news-commentary-v13* data for German to English (DE-EN) task. This training set has $284k$

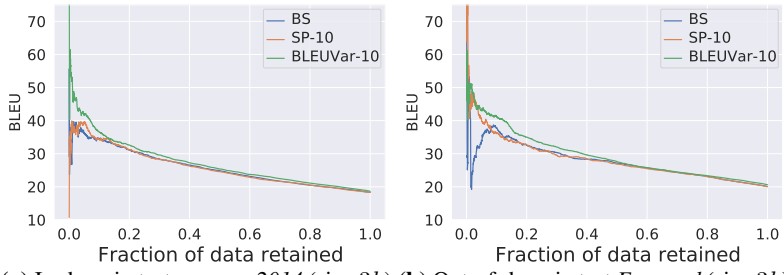

**(a)** In-domain test *newstest2014* (size $3k$) **(b)** Out-of-domain test *Europarl* (size $3k$)

**Figure 1:** Uncertainty measure comparisons using the same-domain test set *newstest2014* (left) and out-of-training-domain test set *Europarl* (right). The Transformer model was trained for DE to EN tasks with the *news-commentary-v13* EN-DE training set (size $284k$) using $350k$ steps.

**Table 1:** Area under a curve (AUC) for all plots in Figure 1.

|  | BS | SP-10 | BLEUVar-10 |
|---|---|---|---|
| (a) Test set *newstest2014* | 25.75 | 25.78 | **27.23** |
| (b) Test set *Europarl* | 27.65 | 28.29 | **29.46** |

samples in the domain of news commentary. During test time, we used both the in-domain test data (*newstest2014*) and the out-of-domain test data (a subset of *Europarl*). *newstest2014* is in the same domain as the training set. *Europarl* contains samples extracted from the proceedings of the European Parliament, which has a different domain than news commentary.

We expect a good uncertainty measure to perform well in both in-domain and out-of-domain test set. As shown in Figure 1 and Table 1, BLEUVar outperforms the other two measures by a large margin for both test sets. In particular, the unstable performance of BS on the out-of-domain test data indicating such uncertainty measure might be not be reliable on out-of-distribution data, i.e. data it never saw before.

### 4.1.2 UNCERTAINTY CAUSED BY DIFFERENT LANGUAGE DOMAINS

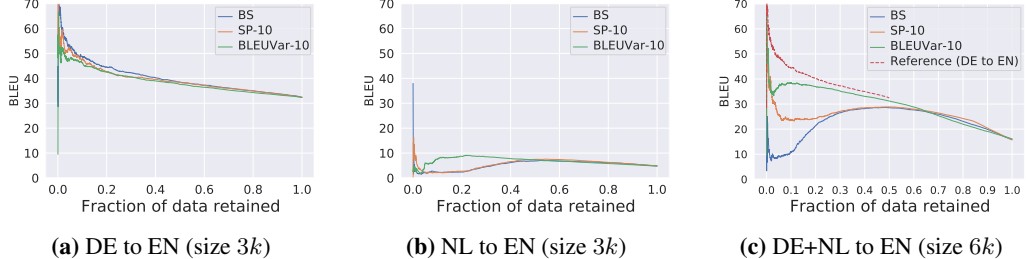

**(a)** DE to EN (size $3k$)      **(b)** NL to EN (size $3k$)      **(c)** DE+NL to EN (size $6k$)

**Figure 2:** Uncertainty measure comparisons using the in-distribution DE-EN test set (a), out-of-distribution NL-EN test set (b) and the combined DE+NL to EN test set (c). The *Reference* line in (c) corresponds to the BS plot from (a), which only has $3k$ test data. Therefore it only reaches the fraction 0.5 in this graph. The model was trained for DE to EN task with the full EN-DE training set (size $4.6m$) using $350k$ steps.

**Table 2:** AUC for all plots in Figure 2.

|  | BS | SP-10 | BLEUVar-10 |
|---|---|---|---|
| (a) DE to EN | **40.22** | 39.43 | 38.61 |
| (b) NL to EN | 5.14 | 5.49 | **6.75** |
| (c) DE+NL to EN | 22.55 | 25.71 | **29.74** |

We trained a Transformer model on WMT13 and Europarl DE (German) to EN (English) sentence pairs (obtaining BLEU 33 on the WMT14 test set). We then evaluated the model on out-of-training-distribution input sentences in NL (Dutch), which shares a large overlapping vocabulary with German (hence input sentences look plausible to non-native speakers). One would hope that such data falling outside of the training distribution would produce model predictions with high uncertainty. A similar experiment with a different language pair (French, German to English) can be found in Appendix B.

The BLEU scores differences between Figure 2(a) and (b) shows that feeding Dutch (NL) sentences into a German to English model does not result in meaningful translations in general. Nevertheless, BLEUVar still provide a better uncertainty estimate with Dutch input (see Figure 2(b), Table 2(b)).

In addition, Figure 2(c) shows the performance-retention curves for the combined DE+NL test set, in which BLEUVar outperforms BS and SP by a large margin. The fact that BLEUVar is close to the DE-EN reference curve indicates most of the DE input has been assigned with high confidence correctly. As the left half of the BLEUVar curve, which corresponds to the most certain half of the test data, nearly resembles the result from Figure 2(a).

A more interesting result is shown in Figure 3. Given the combined DE+NL test set, BLEUVar is able to nicely separate the test set into to two clusters (DE and NL) using only the uncertainty estimates without evaluating on the target translation.

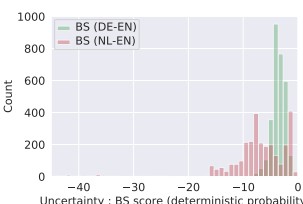 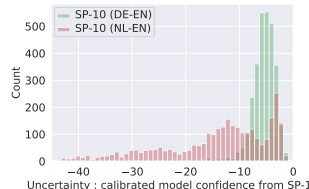 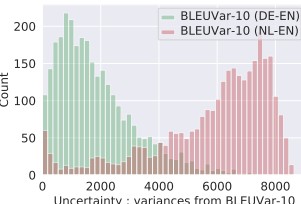

**(a)** BS histogram on a mixed test set (Left corresponds to higher uncertainty)

**(b)** SP-10 histogram on a mixed test set (Left corresponds to higher uncertainty)

**(c)** BLEUVar-10 histogram[3] on a mixed test set (Right corresponds to higher uncertainty)

**Figure 3:** Histograms show uncertainty value for DE-EN (green) and NL-EN (red). Note how BLEUVar-10 is able to clearly separate the in-distribution (green) from out-of-distribution (red). The model was trained for DE to EN task with the full EN-DE training set (size $4.6m$) using $350k$ steps.

Note that we do not try to solve the problem of language detection here. Instead, we use it as an extreme example to show that our uncertainty metric is able to assign very high uncertainty scores towards out-of-distribution test data while being more confident towards in-distribution data. But our methodology was not trained specifically nor designed for language detection. However, being able to separate the two shows that our methods do reflect the uncertainty of the model very well, it also shows that Bayesian deep NLP models can be very powerful.

## 4.2 ANALYSIS OF SENTENCE LENGTH VERSUS UNCERTAINTY

**Table 3:** Average BLEUVar for output sentences of various lengths from Figure 2.

| Lengths | 1-10 | 11-20 | 21-30 | 31-40 | 41-50 | 51+ |
|---|---|---|---|---|---|---|
| DE-EN (In-dist) | 2579.93 | 1867.11 | 1613.95 | 1507.85 | 1502.78 | 2794.11 |
| NL-EN (OOD) | 4772.16 | 5388.25 | 5579.82 | 6039.02 | 6705.95 | 7042.69 |

Use sentences with different lengths and uncertainties from section §4.1.2 as examples, we can see from the Table 3 that long sentences do not necessarily have consistently low/high uncertainty (BLEUVar). For the in-distribution test data, the model is very certain for sentences of length around 21-50, this is because the training sentences length distribution is centred around these sentence lengths (with 54% of the training data in this interval). For the OOD test data, the model is very uncertain across all sentence lengths, and as the sentences get longer the model becomes more uncertain. Note that average uncertainty for the shortest OOD sentences is ∼4700, which is much greater than the average uncertainty for the longest in-distribution sentences, ∼2700.

An important point is that it is not necessary that *all* the short sentences must have low uncertainty, even for sentences in-distribution. I.e., our uncertainty metric is not simply a measure of sentence lengths (or even correlated with it).

We further extended the experiment design and added a naive baseline which simply looks at the sentence length to reject sentences (ordering test data using $\frac{sentence\_length}{longest\_sentence\_length}$, i.e. from short sentences to long sentences). Calculating BLEU scores at different retention rates (as in Figure 2(a), see Table 4), we see that the curve is much lower than BLEUVar. In fact, the performance of sentence length referral is worse than another naive baseline: randomly referring sentences without looking at them at all.

---

[3]With the common practice of BLEU (i.e. $\times 100$), the BLEUVar value results in $\times 100^2$ in the plots.

**Table 4:** BLEU scores at different retention rates under three ordering (sentence length here is system output sentence length; source sentence length behaves the same).

| Retention | Sentence length | Random referral | BLEUVar |
|---|---|---|---|
| 0.2 | 29.62 | 32.06 | 42.87 |
| 0.3 | 31.48 | 31.93 | 41.28 |
| 0.4 | 32.38 | 32.26 | 39.39 |

### 4.3 Introspection into the Model Uncertainty from the Linguistic Perspective

Based on experiments in section §4.1.2, below are some example sentence translations sampled from the model (and with which we estimate the model uncertainty). As a reminder, the uncertainty here (BLEUVar) is the level of disagreement between sampled sentences, as determined by pairwise-BLEU (pairwise between each model output and the other model outputs). For a very confident translation from an in-distribution input (i.e. German), refer to Appendix D.1 Table 10. We can see that all translations sampled from the model are consistent with each other, and the model has no uncertainty at all. For other less certain in-distribution translations as showed in in Appendix D.2 Table 11, we can see that each translation sampled from the model is inconsistent with the others in subtle ways, leading to a larger variability in pair-wise BLEU scores. The model has high uncertainty, but still lower than that of the average OOD sentence. In contrast, Table 5 shows 3 truncated samples from OOD input (i.e. Dutch), the complete table with 5 full samples can be found in Appendix D.3. Here each translation sampled from the model is wildly inconsistent with the others, with some translations reminiscent of nonsense translations often encountered with neural systems when these are run on inputs they never saw before. We can identify these bad translations by the large variability in pair-wise BLEU scores. The model has much higher uncertainty than that of the average in-distribution sentences.

**Table 5:** Out-of-distribution NL source sentence from the experiment in Figure 2(c).

**Source sentence** (NL) :

De debiteurenlanden zouden hun concurrentiekracht terugkrijgen; hun schulden zouden in reële termen afnemen; de dreiging van staatsbankroeten zou - met de ECB onder hun controle - verdwijnen, en hun leenkosten zouden dalen naar een niveau dat vergelijkbaar is met dat van het Verenigd Koninkrijk.

**Reference translation** (EN) : (only used to compute "BLEU to reference")

Debtor countries would regain their competitiveness; their debt would diminish in real terms; and, with the ECB under their control, the threat of default would disappear and their borrowing costs would fall to levels comparable to that in the United Kingdom.

**Model predictive-mean translation** (EN) : (averaging over predictive probabilities during decoding)

The debitenlands were to compete with the rivalrivalrivalrivalrivalrivalrivalrivals of terugkrijgen; they were in debt in the countries of afafafafafafafafafafafafafafafafafafafafafafafafaf

| **Translation "BLEU to reference"** : | 1.9 |
|---|---|

| **Translation uncertainty** : | 8617 |
|---|---|

**Translations sampled from the model**: (3 shorten samples from predictive probabilities during decoding)

| 1 | The debitenlands were the ones to compete in their rivalrivalrivalrivalrivalrivalrivals of them; they were debt-denominated in their afafafafafafafafen; the tripthirthirthirwent of state bankrbankrbankrbankru - with the ECB in its control the run - the run-off - the run - the run-run run run run run of the ECB. |
|---|---|
| 2 | In the debdebdebdebdebits were competitive in terms of law; those debt owed in debt in debt; the three of state bankrbankrbankr - with the ECB, in its, in its, in its, in its control - business - business - the ECB, in its, in the control - the dispute, the - business - the dispute, the dispute, |
| 3 | At the time of its independence, it was a rivalrivalrivalrivalrivalrivaleach one; the debts of the poor; the three-three of the bankrbankrbankrall - with the ECB in its control of the ones - the ones in question - the parties in question, the countries in the future; the three of the bankrbankrbankrbankrbankr |

## 5 Future Directions

With the new tools above we can now develop NMT systems which can be deployed in scenarios where high trust is required of the system, for example in legal applications. With the new tools proposed we can integrate expert annotation in the deployment phase of the system by referring uncertain sentences to human annotation instead of automatic one. Further, with these new tools future research could examine human-in-the-loop approaches to NLP. Such approaches will allow us to develop NLP tools in scenarios when hand-labelling of data is too costly, for example language with scarce resources.

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

## A    IN-DISTRIBUTION EXPERIMENTS

This section details our experiments on data that lays within the training distribution for the WMT English (EN) → German (DE) and English (EN) → Vietnamese (VI) tasks. We explore the calibration of Transformer models in this setting and evaluate the effectiveness of using MC Dropout and the proposed methods to measure the model uncertainty.

In addition to WMT13 dataset for EN → DE tasks mentioned in the previous section, we use the **IWSLT 2015** dataset for translation tasks from EN to VI. There are $133k$ sentences pairs in the **IWSLT 2015** training set and $1.3k$ sentences pairs in the **IWSLT 2015** test set. Both the training and test data for **IWSLT 2015** come from the domain of TED talks.

### A.1    EVALUATING MODEL CALIBRATION

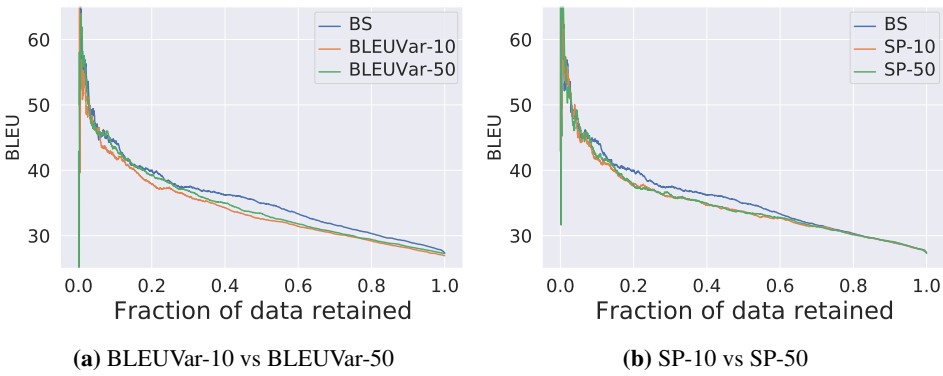

**(a)** BLEUVar-10 vs BLEUVar-50          **(b)** SP-10 vs SP-50

**Figure 4:** Uncertainty estimator comparisons for different number of samples. The model is trained for EN to DE tasks with $4.6m$ training data using $350k$ steps.

**Table 6:** AUC for plots in Figure 4 and Figure 5.

| BS | SP-10 | SP-50 | BLEUVar-10 | BLEUVar-50 |
|---|---|---|---|---|
| 35.78 | 34.86 | 34.93 | 34.14 | 34.68 |

The first question we hope to answer is the quality of calibration in Transformers models and to evaluate the effectiveness of MC Dropout in improving uncertainty estimates.

The Transformer was trained on the full EN-DE training set (4.6 million samples) for $350k$ steps. We evaluate on the *newstest2014* test set.

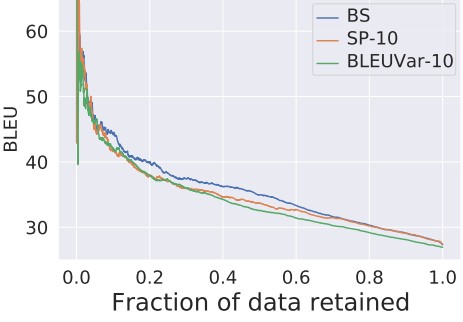

**Figure 5:** BLEU scores for different uncertainty estimators under various retained data rates. The model is trained for EN to DE tasks with $4.6m$ training data using $350k$ steps.

The results from Figures 4 and 5 suggest that the beam search score provides a well-calibrated uncertainty metric on the in-distribution test data. The second observation is that MC Dropout-based methods seem to slightly under-perform beam score in this setting (see Table 6), even when the

number of samples is increased fivefold. In this setting, our proposed metric (BLEUVar) benefits more from increasing the number of dropout samples relative to sequence probability.

## A.2 THE IMPACT OF TRAINING SET SIZE

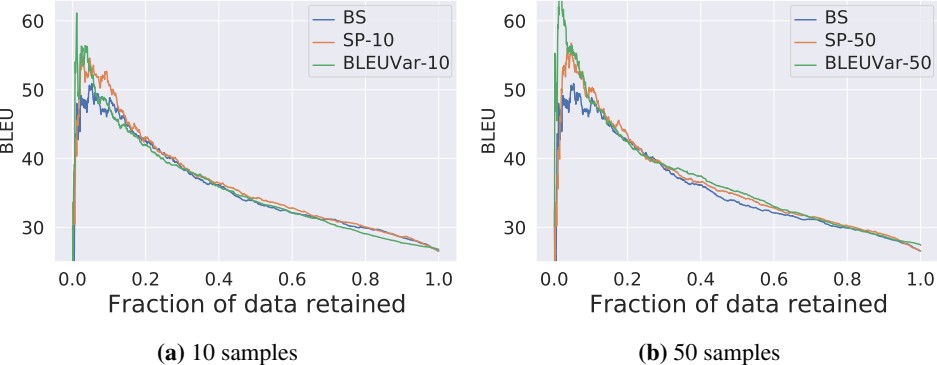

**(a)** 10 samples           **(b)** 50 samples

**Figure 6:** BLEU scores for different uncertainty estimators under various retained data rates. The model is trained for EN to VI tasks with $133k$ training data using $350k$ steps.

**Table 7:** AUC for plots in Figure 6.

| BS | SP-10 | SP-50 | BLEUVar-10 | BLEUVar-50 |
|-------|-------|-------|------------|------------|
| 35.55 | 36.25 | 36.33 | 35.66 | **36.92** |

The WMT EN-DE training set is fairly large and one would assume that most test sentences (or very similar ones) have been observed during training time. Hence we do not expect much epistemic uncertainty to exist in this testing scenario, which the experiments seem to confirm. A natural question to ask is on the effect of training set size on the calibration of models. We explore this question by considering the WMT English to Vietnamese (EN-VI) task which has $133k$ samples in the training set (approx. 2.6% of EN-DE), and down-sampling the EN-DE training set to $50k$ and $100k$ samples.

The performance-retention plots in Figure 6 and the AUC in Table 7 indicate that, while a large training set yields curves that seem to suggest beam score is a sufficient uncertainty metric, when a small dataset is used the MC Dropout-based uncertainty metrics begin to outperform the beam score (note the retention range 0.0 to 0.2). Moreover, in the small training set setting increasing the number of samples drawn from MC dropout results in a significant improvement for BLEUVar.

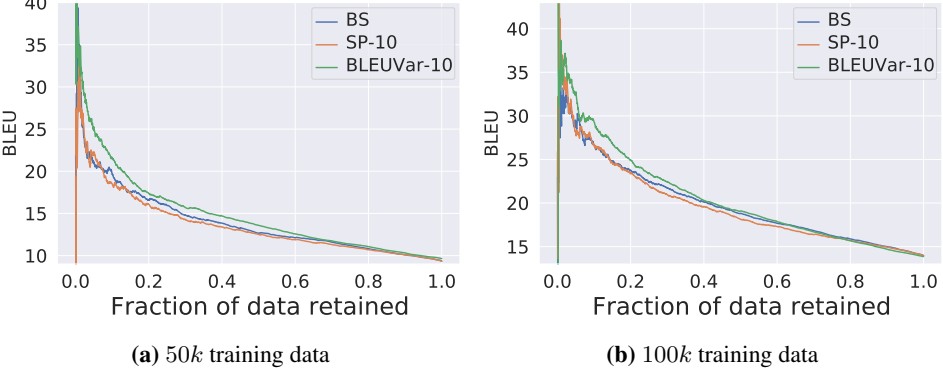

**(a)** $50k$ training data           **(b)** $100k$ training data

**Figure 7:** Uncertainty estimator comparisons for models with different sizes of training set. The models were trained for EN to DE tasks with $50k$ and $100k$ training data using $350k$ steps.

The experiments depicted in Figures 7 and 8 consist of down-sampling the EN-DE training set. Figure 7 and Table 8 demonstrates a similar pattern to the above EN-VI experiment when down-sampling the EN-DE data to $50k$ and $100k$ examples. Again, in the low-data regime BLEUVar substantially out-performs beam score and sequence probability. Figure 8 demonstrates the impact of data size

**Table 8:** AUC for plots in Figure 7.

|                          | BS    | SP-10 | BLEUVar-10 |
|--------------------------|-------|-------|------------|
| (a) $50k$ training data  | 13.97 | 13.61 | **14.89**  |
| (b) $100k$ training data | 19.88 | 19.64 | **20.44**  |

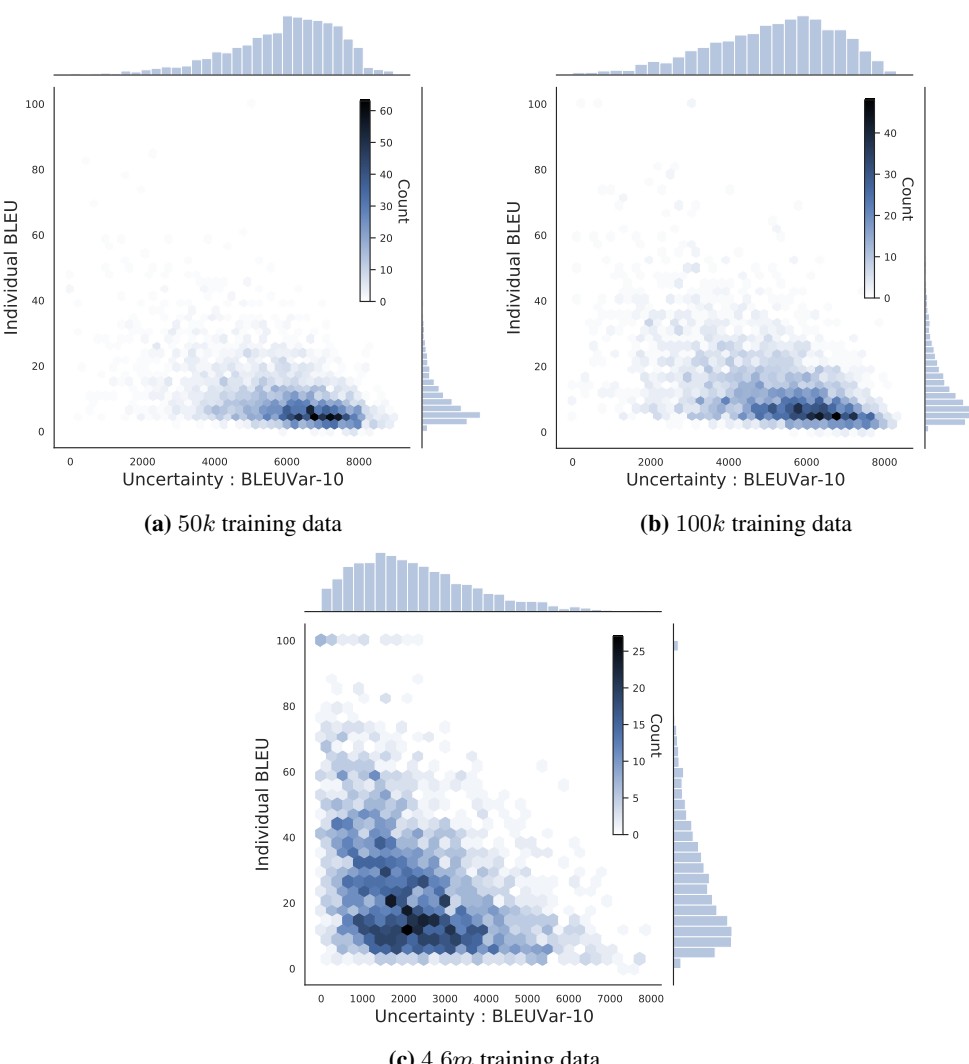

**(a)** $50k$ training data

**(b)** $100k$ training data

**(c)** $4.6m$ training data

**Figure 8:** The density of individual BLEU score versus uncertainty (BLEUVar-10) for all sentences in the same test set *newstest2014* produced by models trained with various size of data set. The sentences are ordered by their uncertainty from low (left) to high (right) using BLEUVar-10. Following the calculation of BLEUVar, since we have 10 samples, the uncertainty estimate BLEUVar-10 has the value in range $[0, 90]$. And we scale it up by $\times 100$, which results in the x-axis has the range $[0, 9000]$. The models were trained for EN to DE tasks with $50k$, $100k$ and $4.6m$ training data using $350k$ steps.

on the distribution of example uncertainty versus performance. We see that low data regimes lead to a low-entropy distribution with high uncertainty across the entire test set; as data availability is increased, uncertainty decreases, and average model performance increases for all rates of data retention.

## B ADDITIONAL OUT-OF-DISTRIBUTION EXPERIMENTS (FR+DE TO EN)

We have done similar experiments as section §4.1.2 on other language pair. Figure 9 here uses a similar experiment design as Figure 2 in §4.1.2. Instead of testing DE and NL on model trained with DE-EN task, here the model is trained with FR (French) to EN (English) task and tests on FR to EN (in-distribution), DE to EN (OOD) and FR+DE to EN test sets. The pair FR and EN has much less overlap in vocabulary than DE and NL.

The BLEU on the full combined test set (see Figure 9(c)) is the best the models can do; then BLEUVar rejects German sentences until it has mostly French sentences left and it has peak performance (data retained=0.3), after which it is forced to reject French sentences as well. Note that performance goes down for small retain rate because there is a small number of DE (OOD) data erroneously being assigned with high confidence, which our metric captures well.

This result is similar to the DE+NL experiments in §4.1.2, with BLEUVar outperforms the rest by a large margin in the mixed test set (see Figure 9(c) and Table 9(c)). Further, the BLEU scores of OOD test is roughly flat compared to the amount of data retained (see Figure 9(b)).

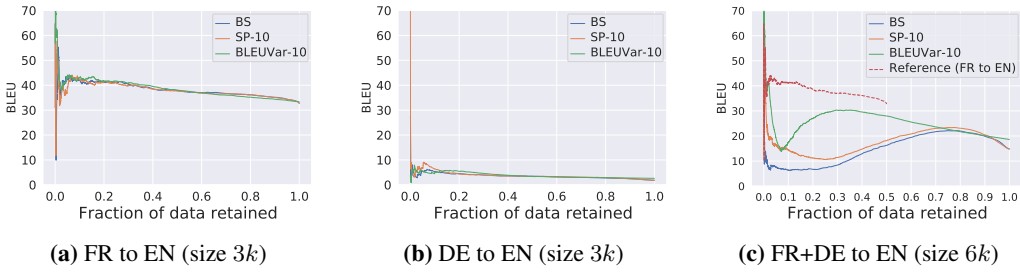

**(a)** FR to EN (size $3k$)      **(b)** DE to EN (size $3k$)      **(c)** FR+DE to EN (size $6k$)

**Figure 9:** Uncertainty measure comparisons using the in-distribution FR-EN test set (a), out-of-distribution DE-EN test set (b) and the combined FR+DE to EN test set (c). The *Reference* line in (c) corresponds to the BS plot from (a), which only has $3k$ test data. Therefore it only reaches the fraction 0.5 in this graph. The model was trained for FR to EN task with the WMT 2014 English-French training set (size $36m$) using $350k$ steps.

**Table 9:** AUC for all plots in Figure 9.

|  | BS | SP-10 | BLEUVar-10 |
|---|---|---|---|
| (a) FR to EN | 38.18 | 38.02 | **38.53** |
| (b) DE to EN | 3.56 | 3.78 | **3.82** |
| (c) FR+DE to EN | 14.70 | 17.67 | **25.10** |

# C    RELATED WORK

Our work might look similar to quality estimation (QE) task in MT (Specia et al., 2010; Blatz et al., 2004), but the problem of QE is fairly different to what we do in this paper. QE assumes the existence of a fixed translation system (e.g., an in-house encoder-decoder attention-based NMT system, as in WMT19's shared task in Quality Estimation). The QE models then have to determine the quality of the system's output. In contrast, we look at the problem of "introspection" where the system has to decide the the confidence ("quality") of *its own* output. This confidence can then be used for selective classification where the model can reject some uncertain translation. Further, standard approaches in QE might assume access to privileged data (e.g., the NMT translations for the source sentences and their corresponding human post-edition, as in task 2 in WMT19'QE), which we do not require. In addition, most existing approaches for QE require additional model to be trained to estimate the translation quality of a MT model, while our method does not have such requirement. Therefore, our method is able to provide uncertainty estimate simply with the parallel corpus used for training the translation model without the need for additional data and training procedure.

The closest to our paper is task 3 in WMT19'QE: a metric to score sentences is sought, which must correlate to human judgement. We would like to stress that a system's confidence in its own prediction does not have to be correlated to human judgement. Indeed, we demonstrate this in Appendix A.2 where a model can indicate that it does not have enough training data, and requires additional data to increase its confidence (the model's subjective view of its uncertainty does not have to correlate with empirical mistakes - Bayesian epistemology (Zalta et al., 1995)).

In addition, QE tasks are mainly focus on estimating the in-distribution translation quality, since the test sets are in the same domain as the training sets provided by WMT QE tasks (e.g. both in the IT domain for English-German WMT18,19). In contrast, the goal for our uncertainty estimate is to identify the out-of-distribution translations, rather then estimating the quality of in-distribution translation. Therefore, our tasks are fundamentally different to QE.

There have been some prior attempts at investigating the similar problem as ours. In particular, Kumar & Sarawagi (2019) investigated the calibration of various NMT models at the token level. Kumar & Sarawagi found that many models are ill-calibrated at the *token level*, leading to the resulting probability distribution over the vocabulary used during decoding is not a good reference for model uncertainty. To correct for this, Kumar & Sarawagi design a recalibration strategy that applies an adaptive temperature to the logits, determined by the token identities, attention entropies, and other relevant components. Desai & Durrett (2020) looked into the calibration of pre-trained Transformers, and discovered that pre-trained Transformers are well calibrated for in-distribution data but ill-calibrated for out-of-distribution data. Such observations on NMT calibration further motivate us to design better uncertainty measures for NMT models.

Another study of uncertainty in NMT models comes from Ott et al. (2018); they found models tend to have overly high uncertainty in their output distribution over sequences. Note that both do not consider *epistemic* uncertainty, not OOD settings. There are some work consider *epistemic* uncertainty (Fomicheva et al., 2020; Wang et al., 2019) and propose MC Dropout-based measures similar to our **Sequence Probability (SP)**. Our work explores this direction and offers a new uncertainty estimation technique (i.e.BLEUVar) that empirically out-performs existing methods by a significant margin.

# D    TRANSLATION SAMPLES

## D.1    IN-DISTRIBUTION CERTAIN SAMPLES

**Table 10:** (Low uncertainty) In-distribution DE source sentence from the experiment in Figure 2(a).

| |
|---|
| **Source sentence** (DE) : |
| Nevada hat bereits ein Pilotprojekt abgeschlossen. |
| **Reference translation** (EN) : (only used to compute "BLEU to reference") |
| Nevada has already completed a pilot. |
| **Model predictive-mean translation** (EN) : (averaging over predictive probabilities during decoding) |
| Nevada has already completed a pilot project. |
| **Translation "BLEU to reference"** :  | 70.7 |
| **Translation uncertainty** :         | 0 |
| **Translations sampled from the model**: (5 samples from predictive probabilities during decoding) |

| | |
|---|---|
| **1** | Nevada has already completed a pilot project. |
| **2** | Nevada has already completed a pilot project. |
| **3** | Nevada has already completed a pilot project. |
| **4** | Nevada has already completed a pilot project. |
| **5** | Nevada has already completed a pilot project. |

## D.2    IN-DISTRIBUTION UNCERTAIN SAMPLES

**Table 11:** (High uncertainty) In-distribution DE source sentence from the experiment in Figure 2(a).

| |
|---|
| **Source sentence** (DE) : |
| Im Grunde genommen sind vegane Gerichte für alle da. |
| **Reference translation** (EN) : (only used to compute "BLEU to reference") |
| Essentially, vegan dishes are for everyone. |
| **Model predictive-mean translation** (EN) : (averaging over predictive probabilities during decoding) |
| Basically vegan dishes are there for everyone. |
| **Translation "BLEU to reference"** : | 34.5 |
| **Translation uncertainty** :        | 3122 |
| **Translations sampled from the model**: (5 samples from predictive probabilities during decoding) |

| | |
|---|---|
| **1** | Basically vegan dishes are for everyone. |
| **2** | Basically, vegan dishes are there for everyone. |
| **3** | Essentially, vegan dishes are available for everyone. |
| **4** | Basically, vegane dishes are there for all. |
| **5** | Basically vegan dishes are there for everyone. |

D.3 OUT-OF-DISTRIBUTION SAMPLES

**Table 12:** Out-of-distribution NL source sentence from the experiment in Figure 2(c).

| **Source sentence** (NL) : |
| --- |
| De debiteurenlanden zouden hun concurrentiekracht terugkrijgen; hun schulden zouden in reële termen afnemen; de dreiging van staatsbankroeten zou - met de ECB onder hun controle - verdwijnen, en hun leenkosten zouden dalen naar een niveau dat vergelijkbaar is met dat van het Verenigd Koninkrijk. |

| **Reference translation** (EN) : (only used to compute "BLEU to reference") |
| --- |
| Debtor countries would regain their competitiveness; their debt would diminish in real terms; and, with the ECB under their control, the threat of default would disappear and their borrowing costs would fall to levels comparable to that in the United Kingdom. |

| **Model predictive-mean translation** (EN) : (averaging over predictive probabilities during decoding) |
| --- |
| The debitenlands were to compete with the rivalrivalrivalrivalrivalrivalrivalrivals of terugkrijgen; they were in debt in the countries of afafafafafafafafafafafafafafafafafafafafafafafafaf afafafafafafafafafafafafafafafafafafafafafafafafafafafafafafafafafafafafafafafafafafafafafafafafafa fafafafafafafafafafafafafafafafafafafafafafafafafafafafafafafafafafafafafafafafafafafafafafafafafaf afafafafafafaf |

| **Translation "BLEU to reference"** : | 1.9 |
| --- | --- |

| **Translation uncertainty** : | 8617 |
| --- | --- |

| **Translations sampled from the model**: (5 samples from predictive probabilities during decoding) |
| --- |
| **1** | The debitenlands were the ones to compete in their rivalrivalrivalrivalrivalrivalrivalrival- rivalrivalrivalrivalrivals of them; they were debt-denominated in their afafafafafafafafen; the tripthirthirthirwent of state bankrbankrbankrbankrbankrbankrbankrbankrbankrbankrbankr- bankrbankrbankrbankru - with the ECB in its control - the run - the run - the run - the run - the run - the run - the run - the run - the run-off - the run - the run-run run run run run of the ECB. |
| **2** | In the debdebdebdebdebdebdebdebdebdebdebdebdebdebdebdebdebdebdebdebdebdebdebdebdebd ebdebdebdebdebdebdebdebdebits, the debdebdebdebdebdebdebdebdebits were competitive in terms of law; those debt owed in debt in debt; the three of state bankrbankrbankrbankrbankrba nkrbankrbankrbankrbankrbankrbankrbankrbankrbankrbankrbankrbankrbankrbankrbankrbankrban krbankrbankrbankrbankrbankrbankrbankrbankrbankrbankrbankrbankrbankrbankrbankrbankrban krbankrbankrbankrbankrbankr - with the ECB, in its, in its, in its, in its control - business - business - the ECB, in its, in the control - the dispute, the - business - the dispute, the dispute, the - business - the dispute, the sovereign |
| **3** | At the time of its independence, it was a rivalrivalrivalrivalrivalrivalrivalrivalrivalrivalrival- rivalrivalrivalrivalrivalrivalrivalrivaleach one; the debts of the poor; the three-three of the bankrbankrbankrall - with the ECB in its control of the ones - the ones in question - the parties in question, the countries in the future; the three of the bankrbankrbankrbankrbankrbankr- bankrbankrbankr ( with the ECB in its control, with the ECB in its control, in the face, the disputes, the financial crises, the financial crisis, the financial crisis, the financial crisis, the financial crisis, the financial crisis, the financial crisis, the financial crisis, the financial crisis, the financial crisis, the financial crisis, the financial crisis, the financial crisis. |
| **4** | De debitenlanden zouden hun concconcurrentierivalrival terugkrijgen; hun levlevlevlevlevlevl ev levlevlevlevlevlevlevlevlevlevlevlevlevlevlevlevlevlevlevlevlevlevlevlevlevlevlevlevlevlevlevl evlevlevlevlevlevlevlevlevlevlevlevlevlevlevlevlevlevlevlevlevlevlevlevlevlevlevlevlevlevlevlevl evlevlevlevlevlevlevlevlevlevlevlevlevlevlevlevlevlevlevlevlevlevlevlevlevlevlevlevlevlevlevlevl evlevlevlevlevlevlevlevlevlevlevlevlevlevlevlevlevlevlevlevlevlevlevlevlevlevlevlevlevlevlevlevl evlevlevlevlevlevlevlevlevlevlevlev |

5 | The debitenland gambgambgambgambgambgambgambgambgambgambgambgambgambgam
bgambgamble in their own countries; the gambgambgambgambgambgambgambgambga
mbgambgambgambgambgambgambgambgambgambgambgambgambgambgambgambgamb
gambgambgambgambgambgambgambgambgambgambgambgambgambgambgambgambga
mbgambgambgambgambgambgambgambgambgambgambgambgambgambgambgambgamb
gambgambgambgambgambgambgambgambgambgambgambgambgambgambgambgambga
mbgambgambgambgambgambgambgambgambgambgambgambgambgambgambgambgamb
gambgambgambgambgambgambgambgambgambgambgambgambgambgambgambgambga
mbgambgambgambgambgambgambgambgambgambgambgambgambgambgambgambgamb
gambgambgambgambgambgambgambgambgambgambgamb

