# OpenReview forum: "Wat zei je? Detecting Out-of-Distribution Translations with Variational Transformers"
_ICLR.cc/2021/Conference — Reject_

### Official Review · AnonReviewer4 · 2020-10-26
**Review for Wat zei je? Detecting Out-of-Distribution Translations with Variational Transformers**

**Rating:** 3
**Confidence:** 4

**Review:**

Summary:

The paper proposes a technique for assessing the uncertainty of a Transformer-based NMT model on a given input $x$. The technique relies on computing a variance-like estimate over a collection of translation candidates for $x$, where these candidates are obtained by perturbing the decoding mechanism through the use of dropout at test time. Experiments compare this technique with other ways of measuring the "epistemic uncertainty" of the NMT model. In limited training data conditions, the proposed measure is better aligned with the actual performance of the model than competing measures, and in particular is better able to detect Out-of-Domain translation requests.

Pros:- The paper addresses an important problem: how to detect translation requests for which a given NMT model is unlikely to produce good results, so that these requests can be handled in a proper way (e.g. referred to a human translator).
- Doing that in a way that only relies on intrinsic properties of the NMT model without requiring a specific quality-estimation training setup, which additionally might have difficulty covering all of the possible out-of-domain cases.
- Proposing to assess uncertainty by an original perturbation mechanism, using dropout at test time, which mimics a form of sampling over the parameter space of the model.
- The clever way in which, in equation (9), a proxy for variance over symbolic target sequences (which is not well-defined) is obtained in terms of the BLEU distances between *pairs* of target sequences.

Issues and Questions:
- The not well-justified relationship (IMO) with Variational Inference, reflected in the title containing the term "Variational", and the introduction of the term "Variational Transformer" to characterise the approach. Section 2.1 on Bayesian Inference and the variational ELBO technique appears to be very loosely connected to the actual technique used in the paper (or at the very least the connection is not made clear). Two issues here: (1) no details at all are provided concerning the training of Transformer with the MC Dropout technique; (2) it is also not clear how fundamental for the approach it actually is to train according to this technique (presumably according to some variational principle ?), as opposed to randomly perturbing the weights of the model at test time in any simpler way --- which would also produce a sample of outputs on which the proposed BLEUVar technique could be applied.
- Related to this point, BLEUVar could also be applied to simpler techniques for producing the target samples than sampling the parameter space, namely simply sampling the output space (hierarchical sampling over the NMT outputs, for the standard fixed parameter value). This would appear to be a rather obvious and relevant baseline. In other words: what is the relative importance in practice of a Bayesian approach to producing different outputs   as compared to the effect of using BLEUVar to measure the variance of candidate outputs ?
- In equation (6) you seem to assume (in the KL term) that $p(\omega) = p(\omega|X)$ (am I correct?), in other words independence of the parameters and $X$. If this is the case, it looks counter-intuitive, because detecting that $x$ is out-of-domain would seem to be very correlated to the structure of $X$, and not only on the structure of $Y$ given $X$ ?
- The experiments do show some superiority of your technique BLEUVar over the non-bayesian BS and the bayesian SP uncertainty measure, in terms of their "correlation" with BLEU, but BLEUVar use BLEU internally, while the other two measures don't, so could this explain its superiority?
- Overall, I did not find the experiments strong. In particular the "different languages" experiments of section 4.1.2 look extremely artificial. You write that German and Dutch have a large overlapping vocabulary and that Dutch sentences "look plausible [German sentences] to a non-native speaker", which is quite an exaggeration. I do understand that your point is of course not to claim that your technique can be used as a practical language guesser, but to show that your technique is able to compute high uncertainty on Dutch sentences (for an NMT system trained on German). However, this experiment can only be seen as a toy experiment (the translations from Dutch can be immediately detected as being ridiculous, on very simple criteria), and more realistic experiments would be needed to convince the reader of the practical applicability of the technique.
- The last section (section 5) states that "With the new tools above we can now develop NMT systems that can be deployed in scenarios where high trust is required of the system, for example in legal applications". Based on my understanding of the paper, this statement appears to be unwarranted at this stage.

------ After rebuttal:
I am lowering my score for the paper. I am not convinced by the responses to several of my questions, in particular to what I felt was an exaggerated insistence on the paper being about  “Variational Transformers”, the rather artificial connection to ELBO (noted by several reviewers), and the lack of self-contained description in the paper of the actual technique used, MC-Dropout, which might be explained in simple and sufficient terms on its own.
Also, I am disappointed that the authors did not update in any way their submission to reflect the reviewers’ comments (contrarily to misleading expressions in the rebuttals). It is therefore impossible to know whether such unconvincing claims as that made in the conclusion “With the new tools above …” would be maintained in the final version.

---

> ### Author Response · Authors · 2020-11-21
> **Response to Reviewer 4**
>
> We would like to thank the reviewer for the valuable comments.
>
> Following are our replies addressing the reviewer's concerns:
>
> > ...  (1) no details at all are provided concerning the training of Transformer with the MC Dropout technique;
>
> As mentioned in the paper, the training procedure of the Transformer remains the same as the original Transformer (_"We trained a Transformer model on ..."_). In fact, any existing pre-trained Transformer model with appropriately tuned dropout layers can be used with our uncertainty measure without retraining or additional training. The only modification is to set the flags for dropout functions to be ‘True’ in the prediction method of the model, and sample multiple times for a given input.
>
> >  (2) it is also not clear how fundamental for the approach it actually is to train according to this technique ...
>
> As demonstrated in Gal 2016, training any neural network with dropout (under certain conditions, e.g. that the dropout probability is tuned appropriately) is mathematically equivalent to optimising a variational lower bound (ELBO) on the model evidence. This is a general result, well accepted within the Bayesian deep learning community, and widely used in fields such as Computer Vision:
> [an example](https://scholar.google.com/scholar?cites=13759824860264424817&as_sdt=2005&sciodt=0,5&hl=en)
>
> This applies to Transformer models as well, a fact we rely on here and explain in our exposition.
>
> This technique allows us to sample from the approximate posterior predictive distribution, and with these samples we can use any “metric” to assess the dispersity of the distribution. Because the samples from the predictive are long sequences of discrete objects, we can’t use off-the-shelf tools like variance to assess how dispersed the distribution is (as explained in our exposition), and instead we propose our metric in this paper, BLEUVar.
>
> >  ... BLEUVar could also be applied to [...] simply sampling the output space ...  What is the relative importance in practice of a Bayesian approach to producing different outputs as compared to the effect of using BLEUVar to measure the variance of candidate outputs?
>
> The Bayesian approach is a standard practice for measuring _model (epistemic)_ uncertainty. Sampling the output space for a fixed model parameter will provide a measure of _aleatoric_ uncertainty. The difference between the two is explained in Kendall and Gal, 2017, but briefly, aleatoric uncertainty measures data ambiguity, e.g. when two labels are equally likely or we have large measurement noise. Epistemic uncertainty measures the model’s _lack of knowledge_, which corresponds to inputs the model never saw at training time. When detecting out-of-distribution we are interested in epistemic uncertainty.
>
> > In equation (6) you seem to assume (in the KL term) that $p(\omega)=p(\omega|X)$ (am I correct?) ...
>
> $p(\omega)$ is not assumed to be $p(\omega|X)$, here $p(\omega)$ is simply the prior of $\omega$ such that $p(Y,\omega|X) = p(Y|X,\omega) \cdot p(\omega)$. $p(Y|X,\omega)$ is our model’s likelihood, and $p(\omega)$ is the prior defined by us. This follows the standard ELBO derivation and we invite the reviewer to go over Gal 2016 if they find this interesting.
>
> >  I did not find the experiments strong. In particular the "different languages" experiments of section 4.1.2 look extremely artificial.
>
> Thank you for the suggestion. This experiment actually was conceived from a discussion about the limitations of Google Translate, which we used as motivation in the text as well: In Google translate, when you input a source sentence in the wrong language it will give nonsense translations (and in recent versions it will suggest an alternative source language). As you correctly point out, our approach could be extended to any language pair, and not just ones where simple rules exist to distinguish the two. But, as stated in the paper, we do not try to solve the problem of language detection here.
> For more realistic experiments, we now supply experiments beyond language pairs, namely comparing news v.s. legal text (Europarl), see __§4.1.1 UNCERTAINTY CAUSED BY DIFFERENT CONTENT DOMAINS__. We will gladly include further experiments from additional domains in a future version of the paper if the reviewer thinks this will make the submission even stronger. But we will not be able to conduct additional experiments in time before the end of the rebuttal period sadly.
>
> ---
>
> We hope that the above clarified and answered your concerns about the paper? We would appreciate it if you could update your score if you feel that your concerns have been met. If not, do let us know if there are any remaining issues - we would gladly answer any outstanding questions or points you felt we misunderstood in the review.

---

> > ### Author Response · Authors · 2020-11-24
> > **Clarification from authors**
> >
> > R1 brought to our attention that we used cumbersome phrasing when referring to the multiple domains experiments in our response - by _"we now supply experiments.."_ we meant that in contrast to our previous workshop version, which did not have these, the ICLR submission includes experiments on multiple domains. We apologise for any potential confusion!

---

### Official Review · AnonReviewer1 · 2020-10-27
**Interesting approach but insufficient comparison with previous work**

**Rating:** 5
**Confidence:** 4

**Review:**

This paper adapts MC-dropout to neural machine translation in order to measure prediction uncertainty to detect OOD samples.

As far as I can tell, the main novelty of this work compared to previous research is the use of BLEU score variance to estimate uncertainty (as opposed to log probability). While the proposed method shows good results when used on out-of-domain data (or when the input data comes from a different language) the paper lacks comparison with previous work. Moreover, the paper's organization relies too much on the appendix. In particular, important discussion of related research is not included in the main text.

Pros:
- The paper is clearly written and well presented
- Good results on out-of-domain data and mixed language data


Cons:
 - Insufficient comparison with previous work. In particular Fomicheva et al. 2020 and Wang et al. 2019 propose similar methods based on the variance of the token level log probabilities (instead of BLEU here). Without these results, it is hard to tell whether the proposed use of BLEU variance is responsible for improvements over the simple "SP" baseline.
 - BLEUVar seems to help more when the data is from a different domain. However, it would be more interesting to see how it fares on a mixture of in-domain and out-of-domain data (after all, this is a more realistic OOD detection setting). This experiment was performed for different languages but I think it would be more important to see it on different domains of the same language (since language identification is easier than domain identification).
 - Too much material is in the appendix: related work, important results on in-domain OOD detection. Results in the appendix are heavily referenced in the main text, which makes it feel like the paper is not "self-contained". The reader has to jump back and forth between main text and appendix to get the whole story.


Remarks:
- "Wat zei je?": Having parts of the title in a different language is fine, especially for an MT paper, but consider adding an English translation in the introduction or as a footnote, as most of the audience does not read Dutch.
- In light of previous work, I suggest toning down grand claims such as "Our new measure of uncertainty solves a major intractability in the naive application of existing approaches on long sentences" in the abstract
- I don't understand the point of mentioning the ELBO. It takes almost half a page but is not used anywhere else in the paper. This valuable space could be used to move more relevant content from the appendix to the main text (such as related work or the experiments in appendix B)
- In section 3, it might be worth mentioning that BLEU (or rather 1-BLEU) is not a proper distance metric. For instance, it is not symmetric, which has important implications (for instance in eq. 12 it is crucial to include both directions)
- Compare the mixed language result with LID? Seems like a relevant baseline for this specific scenario (and more practical than running a neural model 10 times).
- Most figures would be more readable with bars (binned by 10% increments, and with error bars for each bin) than the currebt line plots
- In table 3: I suggest reporting the square root of BLEUVar instead so the scores have the same "unit" as BLEU score (ie. "BLEUStd")
- Typo in 4.2: "Use" -> "Using"
- The repeated references to the appendix made Section 4.3 in particular very hard to read. I suggest the authors focus on discussing mainly the results they can include in the main text, and only refer to the appendix shortly at the end (eg. "see appendix [..] for examples of other cases")

---
Post rebuttal: The authors have partially addressed my concerns with regards to experiments on actual domains. I think this is a central part of the paper and these experiments could be improved, however I am willing to augment my score to 5. I am still ambivalent about the paper but I wouldn't fight against it being accepted.

---

> ### Author Response · Authors · 2020-11-21
> **Response to Reviewer 1**
>
> We would like to thank the reviewer for the valuable comments.
>
> Following are our replies addressing the reviewer's concerns:
>
> >  Fomicheva et al. 2020 and Wang et al. 2019 propose similar methods based on the variance of the token level log probabilities ...
>
> Fomicheva et al. 2020 report that their best results e.g. on Group II in their experiments are achieved by the D-TP metrics. This metric is identical to our baseline Sequence Probability (SP) (with the difference that SP takes the log of the probability and normalises by sentence length, which we explain in the main text). Note that the paper is already cited in our submission, but we will make this connection more clear in the text. We would further like to highlight though that Fomicheva et al. 2020 can be considered contemporaneous to this submission (per ICLR reviewer guidelines, contemporaneous works are ones put online from August 1st - the paper was put online 10 days prior to that).
>
> Wang et al. 2019 seems very interesting as well and is already cited in our submission. However note that they try to solve a different problem (albeit using related tools). They look at the problem of training NMT under small data conditions, and use model confidence to cope with back-translation errors as a tool towards that goal. We will make this point more clear in the text.
>
> >  ...  it would be more interesting to see how it fares on a mixture of in-domain and out-of-domain data ... This experiment was performed for different languages but I think it would be more important to see it on different domains of the same language ...
>
> Thank you for the great suggestion. We now supply experiments on different domains - news v.s. legal text (Europarl), see __§4.1.1 UNCERTAINTY CAUSED BY DIFFERENT CONTENT DOMAINS__. We would gladly include additional domains if you have more suggestions, but will not be able to conduct more experiments in time before the end of the rebuttal period.
>
> > ... reader has to jump back and forth between main text and appendix to get the whole story.
>
> Sorry for the troubles, sadly, we could not fit all the results in the main text due to space constraints.
>
> > "Wat zei je?": Having parts of the title in a different language is fine, especially for an MT paper, but consider adding an English translation in the introduction or as a footnote ...
>
> Great suggestion!
>
> >  ... I suggest toning down grand claims [...] in the abstract.
>
> We would comment that even existing works mentioned above do not solve the intractability discussed in our paper, and instead use the naive approximation which is equivalent to our baseline SP (Fomicheva et al. 2020). We understand the reviewer’s concern though and will tone down the language. These claims are remnants from our original workshop paper from 2019 which this paper extends on.
>
> > ... the point of mentioning the ELBO. It takes almost half a page but is not used anywhere else in the paper ...
>
> This followed earlier feedback we received about the theoretical grounding of the method. The ELBO discussion is used as justification for our approximate inference technique and thus for our use of MC dropout to obtain uncertainty. We will clarify this in the exposition.
>
> > ... it might be worth mentioning that BLEU (or rather 1-BLEU) is not a proper distance metric ...
>
> Will do!
>
> > Compare the mixed language result with LID? ...
>
> We appreciate the suggestion, but as stated in the paper, we do not try to solve the problem of language detection here. Instead, we use it as an extreme example to show that our uncertainty metric is able to assign very high uncertainty scores towards out-of-distribution test data while being more confident towards in-distribution data. Our methodology was not trained specifically nor designed for language detection. However, being able to separate the two shows that our methods do reflect the uncertainty of the model very well, it also shows that Bayesian deep NLP models can be very powerful.
>
> Also, thank you for the advice about the figures, appendix and listing the typos, we will clarify them in the future version.
>
> ---
>
> We hope that the above clarified and answered your concerns about the paper? We would appreciate it if you could update your score if you feel that your concerns have been met. If not, do let us know if there are any remaining issues - we would gladly answer any outstanding questions or points you felt we misunderstood in the review.

---

> > ### Comment · AnonReviewer1 · 2020-11-24
> > **Response to rebuttal**
> >
> > I thank the authors for taking the time to address my concerns.
> >
> > One of the central points I made (as well as other reviewers) were the lack of experiments with multiple domain (and not simply different language), a point which was addressed by the authors with additional experiments. I am willing to increase my score to reflect this.
> >
> > That being said, I feel this particular aspect is not really addressed to the fullest extent. In particular the two domains are rather similar (as evidenced by how close the AUC are, perhaps reporting absolute BLEU score as well would help interpret this). I think experimenting with domains that are more distinct from each others such as news commentary and chat (as mentioned by another reviewer) or social media text (http://www.statmt.org/wmt19/robustness.html) would be a more appropriate experiment. Also, the highlight (to me) of 4.1.2 is the AUC on DE+NL and something similar for multiple domains would be better.
> >
> > Finally, here are some comments on the rebuttal
> >
> > > Fomicheva et al. 2020 and Wang et al. 2019
> >
> > I had noticed that these two papers were cited, which is why I brought them up in the first place. The point on Fomicheva wrt being contemporary is well taken. It is still somewhat unclear to me why there shouldn't be a comparison with Wang et al. or if it is exactly equivalent to one of the baselines.
> >
> > > point of mentioning the ELBO
> >
> > I still don't see the point of Section 2.2 specifically, and I'm not sure I would agree it helps ground the method theoretically (it seems to me that 3.1 is sufficient). To be clear, this is not a deal breaker, but I think my comments with regards to important experiments and related work being in the appendix could be partially addressed by removing this particular sub-section

---

> > > ### Comment · AnonReviewer1 · 2020-11-24
> > > **Additional experiments in revision?**
> > >
> > > It's been mentioned to me that the experiments in 4.1.1 were actually already present in the paper, and in fact after taking another look I can't find any difference between 4.1.1 in the original and current version...
> > >
> > > Did the authors perhaps not upload the revision? If not, the phrasing of the rebuttal "We now supply experiments on different domains - news v.s. legal text (Europarl)" is misleading and I have to reconsider my response.

---

> > > > ### Author Response · Authors · 2020-11-24
> > > > **Clarification and response to Reviewer 1**
> > > >
> > > > We apologise for the cumbersome phrasing - by _"We now supply experiments.."_ we meant that in contrast to our previous workshop version, which did not have these, the ICLR submission includes experiments on multiple domains. Still, we are happy that you found these helpful, and thank you for the response to our rebuttal.
> > > >
> > > > > "I think experimenting with domains that are more distinct from each others such as news commentary and chat or social media text would be a more appropriate experiment"
> > > >
> > > > These are great suggestions! As we mentioned in the rebuttal, today is the last day for us to update the results, but we will add the additional domains you suggested when we can update the draft again.
> > > >
> > > > > "comparison with Wang et al"
> > > >
> > > > In Wang's terminology, EXP is the same as our SP baseline. In their ablation study before their back-translation experiments (their Table 1), Wang study the effect of using several measures of uncertainty as penalties during training, and evaluate their effects on BLEU for translation in-distribution on their Chinese-English development set (which follows the same distribution as their training set). More specifically, they train a translation model with the uncertainty measure used as a penalty on the back-translation (their Eq. 13), and show that their EXP and PTP both lead to worse BLEU. They show that their VAR (variance of predictive probabilities) and CEV (somewhat like UCB) improve BLEU on the development set. They never evaluate whether their measures of uncertainty actually capture uncertainty though. Beyond that, their models require an adaptation of the training procedure, and they don't look at OOD settings. We experimented in the past with VAR on OOD data and found it to not improve over SP, but those results were not included in our final ICLR submission. We will add it back together with CEV as baselines, but would like to stress that if they improve over BLEUVar then we would see this as a positive result, since our main contribution is demonstrating how to perform OOD detection in NMT.
> > > >
> > > > > "removing this particular sub-section"
> > > >
> > > > Following your suggestions, we have decided to shorten this section and add the additional experiments in its place.

---

### Official Review · AnonReviewer2 · 2020-10-28
**The experiments are unrealistic**

**Rating:** 5
**Confidence:** 4

**Review:**

The paper proposed a Baysian method for detecting out of distribution (OOD) in machine translation. To this end, the paper introduces BLEU variance (BLEUVar) that is computed based on a number of samples from Transformer with MC Dropout. The advantage of BLEUVar is that it doesn’t require reference, instead it’s computed based on pairwise comparison of the decoded sentences.
The proposed BLEUVar requires decoding $n=10$ translations of an input sentence. This is not desirable for MT especially each target sentence is decoded conditioning on a particular dropout mask, so it’s not parallelable. I wonder what would be the advantage of this approach compared to a simpler approach of detecting OOD just from input sentences without running the whole translation pipeline.

For OOD experiments, I find that the experiment in section 4.4.1 is a bit unrealistic. For MT, there are many datasets for different domains that can be used as testset (e.g., [biomedical](http://www.statmt.org/wmt20/biomedical-translation-task.html), [chat](http://www.statmt.org/wmt20/chat-task.html), ...). Moreover, the training data in 4.4.1 is too small compared to typical MT training data.

Similarly, I also felt that the experiment on different language domains in 4.4.2 is unrealistic. While the authors applied their method for an extreme case for demonstration. I think there are many extreme domains in translation which are more realistic. It would be nice to see the proposed method applied for those domains (i.e., biomedical, law, IT,...)

---

> ### Author Response · Authors · 2020-11-20
> **Response to Reviewer 2**
>
> We would like to thank the reviewer for the valuable comments.
>
> Following are our replies addressing the reviewer's concerns:
>
> >  ... requires decoding n=10 translations of an input sentence. This is not desirable for MT especially each target sentence is decoded conditioning on a particular dropout mask, so it’s not parallelable ... what would be the advantage of this approach compared to a simpler approach of detecting OOD just from input sentences without running the whole translation pipeline.
>
> A naive implementation would indeed sequentially generate 10 translations one after the other, making translation time 10x longer. However in practice this can be implemented by loading a Transformer batch with the same source sentence duplicated multiple times. Each source sentence replica will be translated with a different dropout mask, since dropout masks are sampled a new for each input in the batch. This is the standard implementation we follow in Computer Vision as well, where this technique is used with large deep models e.g. for uncertainty estimation in medical imaging.
>
> Moreover, one desirable feature for our method is that for existing trained Transformer models which use dropout, with simple change in the prediction function (i.e. turn dropout flag on), BLEUVar can be used almost trivially to provide the ability to measure the model uncertainty without the requirement of retraining the model (which can have significant costs). This is unlike alternative approaches for OOD detection, which will be required in addition to the already-trained Transformer models. As a quick note on OOD detection vs epistemic uncertainty estimation (our approach): OOD detection does not assume a downstream task. So OOD detection does not take advantage of e.g. symmetries encoded in the downstream task model. As an illustrative example, if a downstream task was to rely on the first character in the sentence alone (e.g., _does the sentence start with “a”_), our epistemic uncertainty based approach (our variational Transformer) would learn to ignore the rest of the sentence and look only at the first character. If it is “a” it will be confident, regardless of the rest of the sentence being in German or in Dutch. If it is in Chinese, it will say “I don’t know”. In contrast, OOD detection would reject all inputs which are in Dutch, even ones starting with “a”. In certain applications you would want the former behaviour, and in others you would want the latter behaviour, so the two are complementary to each other. Do let us know if this is not clear enough - we will try to clarify this further, and will incorporate this into the paper as well. As you pointed out, this is an important distinction!
>
> >  ...  experiment in section [4.1.1. and 4.1.2] is a bit unrealistic ... would be nice to see the proposed method applied for domains (i.e., biomedical, law, IT,...)
>
> These are great suggestions! We supplied illustrative experiments that we felt can demonstrate the mechanism at play (similar to when you input a sentence into Google Translate in the wrong source language, and get nonsense as a translation).
>
> We also added examples of news v.s. legal text (Europarl), see __§4.1.1 UNCERTAINTY CAUSED BY DIFFERENT CONTENT DOMAINS__. Indeed there are many more places where our contributions could be applied and demonstrated (further showing the significance of the problem at hand). Sadly we will not be able to conduct any additional experiments in time before the end of the rebuttal period, but would gladly include more experiments on various domains in an updated version of the paper.
>
> ---
>
> We hope that the above clarified and answered your concerns about the paper? We would appreciate it if you could update your score if you feel that your concerns have been met. If not, do let us know if there are any remaining issues - we would gladly answer any outstanding questions or points you felt we misunderstood in the review.

---

> > ### Author Response · Authors · 2020-11-24
> > **Clarification from authors**
> >
> > R1 brought to our attention that we used cumbersome phrasing when referring to the multiple domains experiments in our response - by _"We also added examples of ..."_ we meant that in contrast to our previous workshop version, which did not have these, the ICLR submission includes experiments on multiple domains. We apologise for any potential confusion!

---

### Official Review · AnonReviewer3 · 2020-10-29
**A simple, interesting method for estimating uncertainty in neural machine translation**

**Rating:** 6
**Confidence:** 4

**Review:**

This paper describes a method for estimating a neural machine translation (NMT) system's uncertainty about its translation of a sentence that has two parts: (1) use MC Dropout as a proxy for integrating out parameters; (2) two uncertainty metrics (probability of translation summing over randomly-sampled parameters and variance in BLEU using randomly-sampled parameters). The baseline method is just to use the probability of the 1-best translation under the MLE parameters. The method is evaluated by measuring the BLEU score of a test set retaining only the most-certain fraction of the sentences.

For in-domain sentences, BLEUVar does the best, and SP is the same as the baseline. For out-of-domain sentences (train on news-commentary, test on Europarl), the baseline does dramatically worse, and both BLEUVar and SP are better. The authors also tried treating Dutch sentences as "out-of-domain" and got similar results (BLEUVar > SP > baseline).

I don’t think I understand the authors’ differentiation of the present task from confidence estimation (e.g., Blatz et al., 2004, http://www.alexkulesza.com/pubs/confest_report04.pdf; Ueffing and Ney, 2007, https://www.aclweb.org/anthology/J07-1003.pdf). The authors write that “by definition QE crucially relies on examples of mistranslations to train the surrogate,” but the features these systems use do not necessarily rely on mistranslations; only the classifier does. The present work avoids building a classifier by evaluating the measure directly using performance-retention curves. That’s fine, of course, but I’m not seeing that the distinction between the present work and previous work on confidence estimation is that sharp. Some of the features from older work could have been used as additional baselines here (e.g., some of the ones on pages 41-42 and 45-46 of the Blatz et al. report cited above).

The paper uses a lot of space (about 1.5 pages) presenting fairly technical (to me) background on Bayesian inference, which turns out to not be needed for understanding the rest of the paper. I would suggest cutting this section down. On the other hand, the paper spends very little space explaining MC Dropout. To make the paper self-contained, I think it would be helpful to at least give a brief review of what MC Dropout is and how to do it.

Sections 4.2 reports some experiments/analysis on the relationship between uncertainty and sentence length. I’m uncertain what the research question here is, and what the reader should learn from this section. Section 4.3 shows one example sentence; again, I’m not sure what question this example is meant to answer.

Would it be valuable to study the relationship between certainty (as measured by your measures) and quality? What would a scatterplot of BLEU vs. BLEUVar or SP look like?

In any case, I think it could be interesting to show more than a few examples, maybe 10+ examples, to give a sense for what makes the model more or less certain.

Overall, I like this method and think that it has value for the MT community. I’m not totally sure that ICLR is the best place for this work to be published; the WMT conference would have been a better fit.

Other comments:

- Why is the baseline method called "Beam Score" when all three methods use beam search? Would it make sense to rename "Beam Score" to "MLE Probability" and "Sequence Probability" to "MAP Probability"?

---

> ### Author Response · Authors · 2020-11-20
> **Response to Reviewer 3  --  (2/2)**
>
> > Why is the baseline method called "Beam Score" when all three methods use beam search? Would it make sense to rename "Beam Score" to "MLE Probability" and "Sequence Probability" to "MAP Probability"?
>
> It is called Beam Score since the score value for this method comes directly from the Beam Search score, while values for the other two measuring scores do not. The Beam Score does come from the weights calculated through MLE during training, so calling it MLE probability makes sense but might not be as descriptive as Beam Score for the method. As for Sequence Probability, it might not make sense to call it MAP probability as MAP normally implies the model weights are calculated through MAP during training, which is not the case here.
>
> ---
>
> We hope that the above clarified and answered your remaining concerns about the paper? If not, do let us know if there are any remaining issues - we would gladly answer any outstanding questions or points you felt we misunderstood in the review.

---

> ### Author Response · Authors · 2020-11-20
> **Response to Reviewer 3  --  (1/2)**
>
> We would like to thank the reviewer for the valuable and positive feedback.
>
> Following are our replies addressing the reviewer's concerns:
>
> > ... distinction between the present work and previous work on confidence estimation [CE]  ...
>
> In the Blatz et al. report page 9, they state that: “...[CE models] are typically evaluated according to how well they separate correct and incorrect examples, and strong CE is additionally evaluated according to the accuracy of its probability estimates.”, which means CE cares about whether the translation is correct or not. But in our case, we are interested in identifying whether a translation is OOD, rather than if a translation is correct or not. Such distinction is reflected by the difference in evaluation methods, where CE makes use of a labelled test corpus $D=\\{(x_i, c_i)\\}_{i=1...n}$ (s.t. $c_i$ is 1 if $x_i$ is correct, otherwise 0) [in the Blatz et al. report page 16-17], but our method is evaluated by the performance versus retention curve and does not require such test corpus (in particular does not require $c_i$).
>
> Of course, one can argue that OOD translations are likely to be ‘incorrect’ and in-dist translations are often ‘correct’. In this case, we can set a BLEUVar threshold in our method and define that sentences with BLEUVar lower than the threshold are ‘correct’, then BLEUVar can be evaluated under the CE test corpus. But as mentioned in the Appendix C, QE tasks mainly focus on estimating the in-distribution translation quality, since the test sets are in the same domain as the training sets provided by WMT QE tasks (e.g. both in the IT domain for English-German WMT18,19). In contrast, the goal for our uncertainty estimate is to identify the out-of-distribution translations, rather than estimating the quality of in-distribution translation. Therefore, our tasks are fundamentally different to QE.
>
> > ... give a brief review of what MC Dropout is and how to do it.
>
> Thank you for the suggestion. Briefly, the idea to do MC Dropout is that for models trained with dropout we can keep the dropout sampling turned on during test time, and draw multiple samples given an input. This technique is formally justified through its connection to approximate inference in Bayesian neural networks, hence our exposition explaining this link (i.e., our suggested methodology is not an ad hoc one, but rather well-grounded). We will clarify this in the introduction.
>
> >  ... what the reader should learn from [section 4.2 and 4.3] ...
>
> In §4.2 we try to show that the OOD test set has average high BLEUVar regardless of the output sentence length. One might wonder whether the high uncertainty in our OOD test set comes from the possibility that they produce longer translations than our in-dist test set. And a model might be more uncertain towards longer sentences.  This section shows evidence  to eliminate such concern. We will clarify this in the text.
>
> §4.3 is to provide an example and show what the samples from MC Dropout look like for an uncertain OOD example. Together with Appendix D, we want to show that a highly uncertain input does result in high BLEUVar and largely distinctive samples; in contrast, in-dist data results in low BLEUVar and mostly similar samples. These examples act as a didactic example to show that BLEUVar is a good measure for model uncertainty. We will clarify this as well.
>
> > ... What would a scatterplot of BLEU vs. BLEUVar or SP look like?
>
> Good question! We have already done that and the results are in Figure 8, which shows BLEU vs. BLEUVar.
>
> > ... to show more than a few examples...
>
> Thank you for the suggestion, we have 5 more examples in the appendix for each case. These are placed in the appendix due to space constraints. Is this what you meant?
>
> > Overall, I like this method and think that it has value for the MT community ... the WMT conference would have been a better fit.
>
> Thank you! We indeed attempted submitting this work to NLP venues, but as you would imagine we encountered confusion about the problem we were trying to solve, with the reviewers suggesting that ML venues would be more appropriate. We see our work as part of the first steps bridging the BDL and NLP communities, which as expected requires explaining concepts that could be very unfamiliar to some readers. We would very much appreciate it if you could lead a discussion with the other reviewers as well about your points above!

---

### Decision · Program_Chairs · 2021-01-07
**Final Decision**

**Decision:**

Reject

**Comment:**

This paper

* adheres to the Bayesian interpretation of MC dropout and applies it to Transformer-based NMT, thus approximately sampling from the NMT model's posterior predictive distribution $Y_*|x_*, \mathcal D$
* as the NMT predictive distribution is over a discrete sample space, the authors compute variance of pairwise comparisons between the translation and other candidate outputs in a beam of likely translations (the authors call this BLEUVar).

Whereas the work is potentially interesting it does not seem ripe for publication. Here are some of the issues I'd like to highlight:

1. OOD detection. Detection in input space seems like a natural baseline. The authors argue that OOD detection in output space takes the downstream task into consideration, but going through the conditional also makes the task considerably more difficult and computationally challenging. Though we appreciate the author's point, we don't see it as a good enough reason to discard OOD detection in input space as a serious alternative.

2. Why BDL? The motivation for Bayesian methods is clear, but BDL can at best *approximate* Bayesian reasoning, thus the question does deserve an answer. The reviewers asked for experiments that demonstrate empirically the relevance of the Bayesian formulation, for example, one reviewer suggested to compute BLEUVar in the frequentist case, and that makes perfect sense. Consider this: $q(\theta)$ likely under-estimates posterior uncertainty, so let's say that $\operatorname{Var}(\theta|\mathcal D)$ is rather small, then BLEUVar as presented is in fact not capturing posterior predictive uncertainty (due to entropy of $Y_*|x_*, \mathcal D$), but rather sampling uncertainty (due to entropy of $Y_*|x_*, \theta$).

3. Unrealistic experiments: we all agreed that the experiments are weak. For example, we do not share the authors' excitement for the results around a foreign language as an example of OOD data point, we see it as an artificially simple case. We also expected more interesting cases of mixed domain data sets (for ideas, check tasks within WMT and IWSLT, as well as resources such as Opus and low-resource language pairs as those in FLORES) and more generally different levels of noise (e.g., synthetic data produced by other translation engines, round-trip/back-translations are very typical in low-resource settings).

Additional remarks/suggestions:
* in my personal view, BLEUVar should *not* be based on biased statistics (beam search introduces all sorts of unknown biases); the pairwise comparison mechanism behind BLEUVar is similar to what MT researchers call minimum Bayes risk decoding (a frequentist criterion for making decisions under uncertainty).
* we do believe the setting explored in this paper *is* related to confidence estimation, and even though I agree with the authors that a direct comparison is not per se needed, CE datasets could still prove useful for evaluation;

Though the paper has been appreciated for it dispenses with quality annotation, for it attempts to quantify estimation uncertainty (or epistemic, if the authors prefer) rather than sampling uncertainty (or aleatoric), and for other technical contributions (such as BLEUVar), we think this paper needs more than subtle/careful positioning, it really needs to acknowledge the relevance of certain alternatives and evaluate against them (BDL need not win every comparison, that's not so much the issue, the issue is that the current picture is too incomplete).

A final (personal) remark. I noticed the exchange regarding the suitability of the paper to an ML (vs NLP) venue. I personally do not think your submission is more or less appropriate to one or the other on the grounds of its technical content. The expert reviews attached suggest enough ideas for improvements, and I would imagine an improved version of the paper having a good chance at any major ML (or NLP) venue.